# Accurate and Fast Deep Learning Dose Prediction for a Preclinical Microbeam Radiation Therapy Study Using Low-Statistics Monte Carlo Simulations

**DOI:** 10.3390/cancers15072137

**Published:** 2023-04-04

**Authors:** Florian Mentzel, Jason Paino, Micah Barnes, Matthew Cameron, Stéphanie Corde, Elette Engels, Kevin Kröninger, Michael Lerch, Olaf Nackenhorst, Anatoly Rosenfeld, Moeava Tehei, Ah Chung Tsoi, Sarah Vogel, Jens Weingarten, Markus Hagenbuchner, Susanna Guatelli

**Affiliations:** 1Department of Physics, TU Dortmund University, D-44227 Dortmund, Germany; 2Centre for Medical Radiation Physics, University of Wollongong, Wollongong, NSW 2500, Australia; 3Imaging and Medical Beamline, Australian Synchrotron, ANSTO, Clayton, VIC 3168, Australia; 4Peter MacCallum Cancer Center, Physical Sciences, Melbourne, VIC 3000, Australia; 5Illawarra Health and Medical Research Institute, University of Wollongong, Wollongong, NSW 2500, Australia; 6Prince of Wales Hospital, Randwick, NSW 2031, Australia; 7School of Computing and Information Technology, University of Wollongong, Wollongong, NSW 2500, Australia

**Keywords:** microbeam radiation therapy, deep learning, dose prediction, Geant4, Monte Carlo simulation, preclinical study

## Abstract

**Simple Summary:**

This work describes the development of a fast and accurate machine learning (ML) 3D U-Net dose engine, trained with Monte Carlo (MC) radiation transport simulations, to calculate the dose in rat patients treated in Microbeam Radiation Therapy (MRT) preclinical studies at the Imaging and Medical Beamline at the Australian Synchrotron. Digital phantoms are created based on CT scans of sixteen rats and are augmented to obtain enough anatomical data. Augmented variations of the digital phantoms are then used to simulate with Geant4 the energy depositions of an MRT beam inside the phantoms with 15% (high-noise) and 2% (low-noise) statistical uncertainty. The high-noise MC simulations are used for ML model training and validation, while the low-noise ones for testing. The results show that the ML dose engine provides a satisfactory dose description in the tumor target and generates the dose maps in less than one second.

**Abstract:**

Microbeam radiation therapy (MRT) utilizes coplanar synchrotron radiation beamlets and is a proposed treatment approach for several tumor diagnoses that currently have poor clinical treatment outcomes, such as gliosarcomas. Monte Carlo (MC) simulations are one of the most used methods at the Imaging and Medical Beamline, Australian Synchrotron to calculate the dose in MRT preclinical studies. The steep dose gradients associated with the 50μm-wide coplanar beamlets present a significant challenge for precise MC simulation of the dose deposition of an MRT irradiation treatment field in a short time frame. The long computation times inhibit the ability to perform dose optimization in treatment planning or apply online image-adaptive radiotherapy techniques to MRT. Much research has been conducted on fast dose estimation methods for clinically available treatments. However, such methods, including GPU Monte Carlo implementations and machine learning (ML) models, are unavailable for novel and emerging cancer radiotherapy options such as MRT. In this work, the successful application of a fast and accurate ML dose prediction model for a preclinical MRT rodent study is presented for the first time. The ML model predicts the peak doses in the path of the microbeams and the valley doses between them, delivered to the tumor target in rat patients. A CT imaging dataset is used to generate digital phantoms for each patient. Augmented variations of the digital phantoms are used to simulate with Geant4 the energy depositions of an MRT beam inside the phantoms with 15% (high-noise) and 2% (low-noise) statistical uncertainty. The high-noise MC simulation data are used to train the ML model to predict the energy depositions in the digital phantoms. The low-noise MC simulations data are used to test the predictive power of the ML model. The predictions of the ML model show an agreement within 3% with low-noise MC simulations for at least 77.6% of all predicted voxels (at least 95.9% of voxels containing tumor) in the case of the valley dose prediction and for at least 93.9% of all predicted voxels (100.0% of voxels containing tumor) in the case of the peak dose prediction. The successful use of high-noise MC simulations for the training, which are much faster to produce, accelerates the production of the training data of the ML model and encourages transfer of the ML model to different treatment modalities for other future applications in novel radiation cancer therapies.

## 1. Introduction

In recent years, an increasing number of studies investigating fast dose predictions for radiotherapy treatment planning with GPU algorithms [1] and deep learning models have been published [2,3,4,5]. However, these publications mostly focus on clinically available treatment methods, such as IMRT [6,7], VMAT [8,9] or proton pencil beam scanning [10]. This results partly from the urge for fast dose prediction models to improve clinical treatment plan optimization capabilities [11], but also from the large available datasets from hospitals and their already delivered treatments (e.g., [12]), facilitating the development of machine learning (ML) models. Such large databases are difficult to obtain for novel and preclinical treatments.

This study presents an accelerated development process suitable for preclinical treatments where available training data are limited. Here, our development process is applied to a novel treatment technique, Microbeam Radiation Therapy (MRT) [13], which presents several additional challenges concerning dose calculation. In addition to the limited available training data, the generation of Monte Carlo (MC) datasets is computationally expensive due to the high statistics required to calculate the dose depositions within a few percentage points of statistical uncertainty, resulting from the 24–100 μm-wide microbeams, spatially fractionated with a pitch of 100–400 μm. These spatially fractionated beams result in high peak dose regions, with comparatively low valley doses in between [14]. Several preclinical studies have shown the potential treatment benefits of MRT for tumors with poor treatment outcomes, such as radioresistant melanoma [15], gliosarcoma and lung carcinoma [16,17,18,19]. It is also understood that maximizing the peak-to-valley dose ratio (PVDR) results in better biological outcomes [20]; thus, accurate estimation of both peak and valley doses is necessary. While this work is focused on the application of the developed ML model and its workflow, articles on the working principles and recent progress of MRT are available in the literature [13,14,21].

For clinically available treatments, delivered treatment plans and computed dose distributions in previous patients can often be used as training data; however, such data does not exist for novel treatments and preclinical studies. Instead, frequently evolving phantom designs and irradiation scenarios require new training data for ML models to be calculated on a semiregular basis. This renders the development of ML models unviable for many novel treatments.

Recent proof-of-concept studies deploying ML models to estimate the doses for MRT [22,23] rely on MC simulations with software tools such as Geant4 [24] to create datasets to train the ML models. Even the fastest existing dose calculation methods for MRT [25] require approximately half an hour for one prediction with adequate statistics, hindering its effective use in treatment plan optimization.

This study shows that dose estimations with satisfactory accuracy can be obtained within milliseconds with the developed ML model, even with a small amount of training data available, by implementing data augmentation techniques and training the models with relatively low-statistics (15% noise) MC data, which are significantly faster to acquire.

The rest of this paper is structured as follows. Section 2 describes the setup of the used MC simulation to generate the dose distribution data. This is followed by a description of the rat CT scans and digital phantoms used in this study. Then, the method to train and test the ML dose engine is presented. Section 3 presents the model optimization results and the application to realistic test patient data. Finally, the findings are discussed and summarized in Section 4 and Section 5.

## 2. Materials and Methods

### 2.1. MC Simulation

In this work, an existing Geant4 simulation [26,27] was adopted to model the generation and transport of synchrotron radiation at the Australian Synchrotron’s Imaging and Medical Beamline [28]. The position, energy and direction of each photon of individual microbeams was recorded in a Phase Space File (PSF) just before entering the target. Then, in the Geant4 simulation developed and used in this work, the PSF is used to describe the incident radiation field on the target and to calculate the associated energy deposition in the treatment target. The advantage of this approach is the ability to use the same PSF to calculate the dose in different targets/anatomies, speeding up the overall MC simulation executions in preclinical research for MRT (e.g., [18]). The simulated microbeam field is rectangular and has a fixed size of 8 × 8 mm${}^{2}$, adopted from the applied treatment protocol of the preclinical study this work is based on [18,27].

The Geant4 prebuilt electromagnetic physics constructor EmStandardPhysics Option 4 [29] is adopted to model the interactions of photons and electrons in the simulation geometry. The effect of polarization on photons processes is considered by using the Livermore models [30]. All MC simulations are performed using Geant4 10.6p02 [24].

### 2.2. Energy Deposition Scoring in Voxels Implemented in the MC Simulation

Figure 1a shows the energy deposition produced by the central 3 mm-wide region of the microbeam field entering a water phantom. The dose is deposited mainly along the tracks of the X-rays (peaks regions). The peaks have a width of few micrometers and are separated by valleys where the dose is significantly lower. In this study, the pitch between two peak is 400 μm. MC simulations developed for MRT dosimetry usually adopt a micrometer-sized voxelization [26,27] to describe the dose with satisfactory spatial resolution; however, this method is computationally not feasible in the scope of this study. Instead, in this work, the energy depositions in the pathway of the peaks (width of the scoring window: 10μm), and the energy depositions in the valleys between them (width of the scoring window: 100μm), are scored separately and then assigned to the respective macrovoxels (represented as white pixels in Figure 1b). This approach allows a macroscopic description of peak and valley doses and is more feasible for ML predictions, as it significantly reduces the number of required voxels in the prediction volume. The PVDR is calculated as the ratio between the dose in the peak and in the valley for each macrovoxel. A volume of 48×8×8 mm${}^{3}$ (depth × width × width) is recorded using this technique with a macrovoxel size of 0.5×0.5×0.5 mm${}^{3}$, resulting in 96×16×16 voxels for each data sample.

For each geometrical configuration (corresponding to an individual rat head phantom), the simulation was repeated twenty times with different random seeds. Then, the results of the repeated simulations were used to calculate the mean value and standard error of the energy deposition in each voxel.

### 2.3. Rat Head Phantoms

The digital phantoms of the rat heads used in this work are based on CT scans of a total of 16 rats, two weeks after implanting 9 L gliosarcoma cells [31] sourced from the European Collection of Cell Cultures (ECCC). The age of the rats is approximately six weeks at cell implantation. The rats are imaged and treated eleven and twelve days post implantation, respectively. The average body weight is 184.5 ± 9.2 g on treatment day [18]. The CT Scanner has a pixel spacing of 0.4–0.6 mm and a slice thickness of 0.6 mm.

The CT scans are used to create rats’ digital phantoms, which are then imported into the MC simulation following the workflow detailed in [27]. In the first step, the centers of the brain of all CTs are manually identified, and the CTs are rotated, resulting in a skull orientation perpendicular to the *X*-axis, which coincides with the beam direction. The Hounsfield Units (HU) from the CT scan are used to manually categorize the phantom voxels into three material classes: air (G4_AIR [32]), water (G4_WATER [32]) and bone (G4_BONE_COMPACT_ICRU [32]). Finally, a 5mm-thick bolus layer (G4_WATER [32]) is placed on top of the rat phantom as per the experimental setup. An example of a digitized rat phantom is shown in Figure 2. More details on the segmentation process are provided in [27].

### 2.4. High-Noise Monte Carlo Simulation Datasets for Training and Validation

Given the limitation that only the CT scans of sixteen rats were available for this study, we use ten scans for the training data to obtain the largest possible variety and three scans each for the validation and test data to obtain a statistically meaningful variation for the performance evaluation during the hyperparameter optimization and for the final unbiased assessment. Rat CT scans were in no particular order in the dataset. Therefore, selecting the first ten for training is equivalent to a random selection of ten samples.

To maximize the available training data from the limited patient CT data, data augmentation is performed to increase the number of samples by artificially generating samples. This is achieved by randomly applying transformations to the digital phantoms before running the MC simulations: translating them (±5 mm up and down from the beam’s view, perpendicular to the beam as far as the beam still targets the brain), rotating around the center of the brain (±10 degrees around each axis) and scaling the size of voxels isotropically in the three dimensions (factor 0.8–1.2). With this method, a total of 6500 simulation data samples are produced: 4569 samples (generated with rats number 1–10) for training, 1431 samples (generated with rats number 11–13) for validation and 500 samples (generated with rats number 14–16) for testing.

Figure 3a shows an example of an energy deposition map in a peak and a valley, in the central plane of the digital phantoms of the training data. The distribution of statistical uncertainty of the voxels, quantified with the standard error, peaks around 15% for the valleys and around 5% for the peaks, as can be seen in Figure 3b. MC simulations with this type of uncertainty are referred to as high-noise in this paper and are used for ML training and validation.

### 2.5. Low-Noise Monte Carlo Simulation Datasets for Testing

In addition to the high-noise test samples, three low-noise test samples were simulated at the actual tumor positions (shown in Figure 4) for test rat number 14, 15 and 16. These samples are used to compare the dose predictions of the ML model with the MC simulations in the whole prediction region but with special attention to the tumor volume for realistic, delivered treatments, without being dominated by statistical noise.

The statistical uncertainties compared with the high-noise datasets are significantly lower in these treatment test samples. Figure 5a shows an exemplary energy deposition simulation in which the lower noise is visible, as less fluctuations between voxel colorization and a smoother outline out of field result from the crop of visualization at 5% of the maximum energy deposition. The histograms in Figure 5b shows that the statistical uncertainty in the peak areas is below 0.5% for 97.6% of the voxels of the low-noise samples (mean value = 0.36%), while they are less than 2% in the valley for 98.1% of the voxels (mean value = 1.23%).

### 2.6. Machine Learning Model

The ML model is the same as in our previously published study [22] and is based on a 3D U-Net [33], illustrated in Figure 6. The models are implemented using Tensorflow v2.2 [34].

The input of the model comprises the 96 × 16 × 16 density matrix of the phantom within the prediction volume, indicated with red lines in the schematic. Both the material density (input) and energy deposition (output) matrices are normalized to the range [0,1] using the respective minimum and maximum values in the training dataset.

A compression path using strided convolution layers with a following decompression path using blocks of 3D upsampling followed by two convolution layers achieves a multiscale extraction of relevant geometric features from the input to subsequently predict the energy deposition in each voxel. Skip connections between the compression and the decompression path allow bypassing deeper model layers, allowing features of each compression level to be used for the prediction and to avoid vanishing gradients [35].

Two independent ML models are trained for the peak and valley energy deposition predictions, respectively. For each of them, an individual search for an optimal hyperparameter configuration is conducted by evaluating the performance on the validation data.

In the scope of this study, the number of convolution filters per convolution layer, the batch size and the learning rate are varied in the optimization. For this, different neural networks with respective settings are trained. For the training, the Adam optimizer [36] is used, together with the mean absolute error (MAE) between the predicted and MC-simulated energy deposition as loss function. The MAE is calculated as $MAE=\frac{\Sigma_{i}^{n}\left|y_{i}-x_{i}\right|}{n}$, where $y_{i}$ and $x_{i}$ are the energy depositions calculated by the ML dose engine and the MC simulation in voxel *i* of the target region, with a total number *n* of voxels. Training is stopped when the MAE computed on the validation data does not improve anymore for at least 30 epochs. The models of the respective epoch which achieve the lowest MAE on the validation data are used for obtaining the predictions for the test cases and for the corresponding comparisons.

### 2.7. Performance Measures

In the search for optimal hyperparameters, the MAE computed on the validation data is used as the main measure of comparison.

Due to the shifts and rotations used for data augmentation, several data samples exhibit clinically less relevant features, such as large proportions of the spine or auditory canal, which are both not subject to MRT treatments under current preclinical protocols at the Australian Synchrotron. Voxels with bone material especially lead to large MAE values, as the energy depositions are larger within bone structures compared with the brain. To allow for a more outlier-robust comparison of the ML model performance on the training, validation and test datasets, not only the average MAE but also the resulting boxplots are analyzed, which contain more information about the distribution of deviations.

Much of the deviation of the ML predictions from high-noise MC data results from statistical fluctuations of the MC simulations themselves. An accurate voxelwise prediction of energy deposition of high-noise MC simulation data is not only not desired but would also mean poor generalization. Instead, an estimate of the mean value of the underlying energy deposition distribution is desired, which would match an MC simulation of the same scenario with less statistical uncertainty. In order to investigate if the ML model is capable to interpolate the high-noise data, the smooth ML energy deposition predictions are compared with low-noise MC simulation data as well as with the high-noise MC simulation data, relative to their standard error.

In the case of an unbiased prediction of the values for each voxel, it is expected that 68% of values lie within one standard deviation (1σ from the simulation mean value). This expectation value can be used to assess the ML prediction quality in the presence of noisy MC data: if less than 68% of voxelwise ML predictions lie within 1σ from the MC simulation, the deviations cannot be explained solely by statistical fluctuations, hinting at a systematic deviation of the ML prediction from the simulation. If, on the other side, more than 68% of the voxelwise energy depositions agree within 1σ between ML and MC models, it points towards an overfitting of the model to the noise present in individual data samples.

In the case of the three test patient cases with low-noise MC simulations used for testing, the relative deviations $\Delta D_{rel}=\frac{D_{ML}-D_{MC}}{D_{ML}}$ between the ML prediction and MC simulations are assessed, where $D_{ML}$ and $D_{MC}$ are the doses predicted with the ML and MC models, respectively, in each voxel of the target. Two-dimensional visualizations of $\Delta D_{rel}$ are mostly shown in discrete steps in the plots. This is done to allow for an easier visual inspection of results by the reader, which is more difficult using continuous color scales. The agreement between MC and ML predictions is deemed satisfactory when $\Delta D_{rel}<\pm$3%. This criterion is chosen in agreement with the commonly used 3% gamma index [37]. However, in contrast to the gamma index, no spatial deviation is allowed, as two computational data samples are compared voxel by voxel.

In addition, the prediction of the biologically important peak-to-valley dose ratio (PVDR, [20]) is compared between ML model and MC simulation.

## 3. Results

### 3.1. Hyperparameter Optimization

The best average MAE on the validation data in dependence on the different hyperparameter settings, together with the corresponding MAE on the training data, are shown for the valley model in Figure 7a and the peak model in Figure 7b. In the prediction of the valley energy deposition, the model with 64 convolution filters in each convolutional layer, a batch size of 8 and a learning rate of $1\times 10^{-3}$ resulted in the best validation performance. Training with smaller batch sizes or larger learning rates did not converge. The best model for the peak predictions used 128 convolutional filters in each convolution layer, a batch size of 8 and a learning rate of $5\times 10^{-3}$. Although training with smaller batch sizes and larger learning rates did converge for these training runs, no better results were achieved.

### 3.2. Performance and Generalization Assessment

The optimal peak and valley ML models are used to predict all high-noise training, validation and test data samples to assess the overall performance and generalization. Figure 8 and Figure 9 show examples of results for the valley and peak regions, respectively. Figure 8a and Figure 9a show boxplots of the MAE for the training, validation and test data. Figure 8b and Figure 9b illustrate depth–energy deposition curves at the center of the microbeam field for one exemplary validation data sample for both MC and ML models. The predictions of the ML model agree well with the simulated data within the statistical uncertainty of the MC simulation, while being significantly smoother, which contributes to the assessment that the model can generalize to unseen test data. Larger deviations can be seen in areas of very low density as occurring in both samples deeper in the phantom. A closer investigation into the generalization and performance is conducted by analyzing the fraction of voxels for which the deviation between the ML and MC prediction is smaller than 1 σ of the statistical uncertainty of the simulation. As shown in Figure 8c and Figure 9c, the distribution of the training data peaks at a value of around 65%, which is the closest to the expected value of 68%, which means the deviations are mostly of pure statistical nature. While the distribution of the validation data is only slightly broader and shifted to lower values, the distribution of the test data averages around 61% and is visibly broader for the peak predictions, which indicates that the model does not generalize perfectly to the unknown geometries of the test data.

The averaged fractions of voxels with deviations smaller than 1 σ are shown together with the averaged MAE for all three datasets in Table 1. While the averaged MAE agrees well within uncertainties between the three datasets, the averaged fraction of voxels indicates a small bias on a voxel-by-voxel basis if the model is evaluated with independent data.

The training loss is observed to be higher than the validation loss, at least for the chosen peak prediction model, and this tendency is visible for multiple valley prediction models in Figure 7a as well. This is a result of simulation samples from the rats with numbers 1 and 8 (both in the training dataset), exhibiting a larger number of samples than average that include a relevant proportion of spine in the path of the beam.

In Figure 10a, an exemplary prediction of a training data sample with a large proportion of bone is shown as a 2D slice and is compared with the MC simulation relative to its statistical uncertainty. The corresponding depth profile, indicated with a red (black) dashed line in the 2D slices of the energy predictions (relative differences), is shown in Figure 10d. Even though this case is not representative of the treatment field used in this preclinical work, the model is capable to predict the energy depositions quite accurately, despite the large gradients in energy. In the bone voxels, there is more energy deposition; therefore, the absolute differences in this physical quantity are larger than those calculated in water, although the relative differences are the same, as shown in Figure 10a. This results in larger MAE for samples comprising a larger number of bone voxels. However, when comparing the performance on the different datasets using boxplots (see Figure 8a and Figure 9a) instead of only the mean MAE value, which is more robust against outliers, the effect of larger absolute differences in the energy deposition in the bone is less significant or not visible at all.

The two examples of the test data with the respective lowest agreement between ML prediction and MC simulation are shown in Figure 10b,c for the peak and valley region, respectively. Figure 10d,e shows one depth–energy deposition curve for each of these samples at a position indicated by red dashed lines in the 2D visualizations. In the case of the peak model, a systematic overestimation of the energy deposition behind an air pocket in the phantom (auditory channel) can be seen. In the case of the valley model, predictions of relatively thin bone structures especially lead to lower agreement with the MC data. Despite the fact that these are extreme cases and clinically irrelevant cases for MRT, the model still does a reasonably satisfactory job in predicting the deposited energies.

### 3.3. Predictions for Test Rat Patients

The energy deposition predictions for the three treatment cases described in Section 2.5 are converted to dose in units of Gray to link the results more directly to their preclinical implications. The target geometries around the treated tumors and the respective peak and valley dose prediction deviation are shown in Figure 11 for an exemplary case (rat number 14, for which the lowest agreement between ML prediction and MC calculation was found) and compared with the low-noise MC simulation data. The figure shows the percentage difference of relative dose and the depth–dose curves for the peak and valley doses at the center of the prediction volume. At least 93.9% of all voxels of the peak dose prediction and at least 77.6% of all voxels of the valley dose prediction exhibit less than 3% dose deviation (see Table 2). Especially in the region of the tumor, indicated with a white overlay in Figure 11, the agreement is very high; a deviation of at most 3% is achieved for at least 95.9% of the valley dose voxels and 100.0% of the peak dose voxels, respectively. Towards the distal end of the phantom, systematic deviations of the ML prediction from the MC simulation can be seen either over- or underestimating the doses, mostly within 10% agreement, which is the case for 98.5% and 97.6% voxels in the peaks and valleys, respectively.

The respective fractions of voxels with a difference in terms of relative dose $\Delta D_{rel}<$ 3% in the full prediction volume, in the tissue volume only and in the tumor volume only are shown in Table 2. The peak predictions especially show agreements within 3% for over 93% of the full prediction volume and 100% for the tumor targets. The valley dose predictions exhibit a larger fraction of deviating voxels, which may be explained by the larger statistical uncertainties of the valley dose MC training data when compared with the peak dose data. Nevertheless, we also obtained an agreement within the tumor volume above 95% for the valley dose.

By predicting both the peak and valley doses, the biologically important PVDR can also be calculated with the ML model (see Section 2.2). As an example, the predicted PVDR for test rat 14 around the treatment site is shown in Figure 12. Comparing the deviations with Figure 11a, it can be seen that the deviations of the valley dose predictions are the main driver of PVDR deviations. In all three test cases, the deviation of the predicted PVDR from MC data is less than 5% for approximately 97% of all voxels averaged over the three rats for peak predictions, and approximately 94% of all voxels for the valley predictions. Figure 12b shows the values and deviations together with the respective statistical uncertainty along the center of the prediction volume.

Using the ML models of the peak and valley regions, it is possible to assess the impact of hypothetical treatment planning using the ML models. During preclinical rat treatment, according to the method defined in [18,27], the irradiation duration is defined by choosing a prescription valley dose $D^{*}$ and exposing the patient to as much irradiation as needed to achieve a minimum valley dose of $D^{*}$ in the entire tumor, obtaining 100% coverage. Using this prescription method, the minimum valley dose predictions using ML and MC are compared to assess the resulting difference in applied dose to the rats. The resulting differences are shown in Table 3 and are at maximum approximately 1%. This means that a treatment plan based on the predictions of the ML model would be acceptably accurate in terms of total delivered dose when compared with MC simulations.

## 4. Discussion

This study presents an essential step in advancing fast dose prediction models for MRT treatment planning. Although trained on relatively high-noise MC data with mean statistical fluctuations of 5% (15%) in the peak (valley) region, the developed ML model exhibits dose deviations of under 3% compared with low-noise MC simulation data for most voxels (>95%) in the tumor volume in the three exemplary treatment cases under study. The prediction of a dose distribution (using a preprocessed density matrix as input) takes approximately 50 ms, which is significantly faster than the currently fastest calculation method, which takes approximately 30 min [25]. Batch processing allows for simultaneous prediction (up to 32 samples with an Nvidia GForce 1080t GPU with 11 GB memory). When compared with high-noise MC simulations, the ML dose engine is approximately one million times faster to predict the peak and valley doses in the macrovoxels. To note, the MC simulations are executed on CPUs, while the ML dose engine on GPUs.

While the achieved dose prediction accuracy is of more than 3% within nearly all of the tumor volume for both the peak (100.0% of voxels) and valley (>95.5% of voxels), its dosimetric performance may be improved outside the target tumor (close to air cavities and bone structures), increasing the training set with a larger number of CT scans from rats. The difference of the ML model performance between the peak and the valley dose predictions may be partly explained by the different noise in the used training MC data. Nevertheless, the ML model predictions agree very well, even when compared with the low-noise MC data. Only around 20% of the voxels of the full phantom, or around 4% of the voxels in the tumor volume, deviate by more than 3%, although the model was trained with on average 5% and 15% noisy data for peaks and valleys, respectively. This gives us confidence that the model generalizes well and learns a very good approximate of the underlying function despite the noise.

While the presented ML training method and the achieved results provide a very promising outlook for future studies, important limitations of the study need to be stated. The ML model is trained only on a single set of beam characteristics, such as energy profile, divergence and fixed MRT field size. In addition, the used target phantoms exhibit only limited variation, as irradiations were performed only from the top of the skull. Another limitation, which probably is common when developing ML dose engines for radiotherapy treatments at the preclinical stage, is the small number of CT scans that are available; therefore, it was necessary to augment the data artificially. Another implication arising from the relatively small number of data points, especially the three independent test subjects, is that individual features of each of them contribute strongly to the final results. While this is notable in the reported results, the performances on the training, validation and test subjects were found to be statistically comparable between those data points. Regarding the presented model trained only on the limited number of CT scans available, we believe that this shows a sufficient degree of generalization. Regarding future studies, the success of the approach should be reproduced and validated with a larger number of test subjects with a larger degree of variation to show the generalization capability of such a model more significantly. When comparing the performances on test and training data, small systematic biases occur, which tend to hinder the accurate prediction of the valley doses around air cavities and bones, especially outside the tumor. Still, despite this limitation, the dose prediction of the ML dose engine is satisfactory and, when applied to a clinical environment, the ML dose engine should generate even better results than shown here, thanks to the availability of a larger training dataset, which would translate to better predictive power of the ML model. Another limitation is that the ML dose engine needs to be adapted and, at least, partially retrained when applying it to other cases (e.g., changing target phantom, filters and magnetic field of the MRT beamline and radiotherapy treatment).

Our results indicate that it is feasible to train ML prediction models with satisfactory accuracy using relatively high-noise MC training data. The clear advantage of using high-noise MC simulations is the acceleration of the training of the ML dose engine when applied to spatially fractionated therapies such as MRT, for which MC simulations are usually very time-consuming. The high-noise MC samples used in this study in the training and validation are acquired with 1/40th of the simulation time of the low-noise MC simulations. The significantly reduced execution times of the high-noise MC training data actually allows for easy adaptation to different preclinical conditions. Investigating the percentage of voxels exhibiting a dose deviation of less than one standard deviation and comparing it with the expectation of 68% allows for a meaningful interpretation of smooth ML predictions. This is more difficult when only using measures such as the MAE, which is mainly driven by the deviation between ML prediction and MC simulation caused by the statistical uncertainty of the data.

An aspect for future studies would be the quantitative investigation of the dependency of the dose prediction accuracy of the ML dose engine on the statistical uncertainty of the training data, which was not investigated in the scope of this study. In this work, the presented high-noise MC simulation data were chosen as exemplary.

In future extensions of the developed ML model, a considerable additional benefit could be achieved by increasing the training set and including larger prediction volumes, especially out-of-field, for a better estimation of doses to organs at risk (OAR) around the tumor, which is currently not considered in preclinical treatments at the Australian Synchrotron.

## 5. Conclusions

This study presents the first successful application of an ML model for MRT dose prediction in a preclinically relevant scenario. The training data comprise high-noise MC data, which are much faster to acquire than low-noise MC data. The resulting ML predictions are smooth and do not exhibit the noise present in the MC data. A comparison with low-noise test data shows that the predicted doses are accurate within 3% for at least 77.6% of all predicted voxels (at least 95.9% of voxels containing tumors) in the case of the valley dose prediction and for at least 93.9% of all predicted voxels (100.0% of voxels containing tumors) in the case of the peak dose prediction. The ML model seems to generalize well, even if we use training MC data with relatively high statistical uncertainty (15%).

The findings of this study allow for an optimistic outlook for the development of ML models to quickly predict doses for preclinical and especially spatially fractionated treatments, which usually require long MC simulation times. Future studies will translate the findings to other MRT treatment settings, including different beam modalities, conformal irradiations and new target phantoms.

## Figures and Tables

**Figure 1 cancers-15-02137-f001:**
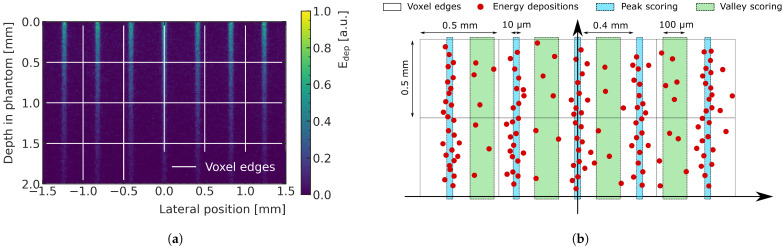
(**a**) Energy deposition in water by a subset of coplanar X-ray microbeams, typical of MRT, entering from the top of the shown region. The dose is deposited mainly along the tracks of the X-rays (peaks regions). The peaks are separated by valleys where the dose is significantly lower. Macrovoxels are shown in white. (**b**) Sketch showing the concept of scoring energy deposition into macrovoxels (represented as white pixels with 0.5 mm lateral sizes). The energy deposition is calculated in the peaks (light blue regions) and in the valleys (green regions) and then associated to the macrovoxel containing it. Adapted from [23].

**Figure 2 cancers-15-02137-f002:**
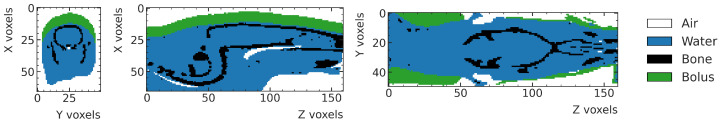
Example of digital rat phantom obtained from the segmentation of CT scans. Defined materials (air, water and bone) are assigned to individual voxels. Green voxels are associated to the bolus, modeled as water.

**Figure 3 cancers-15-02137-f003:**
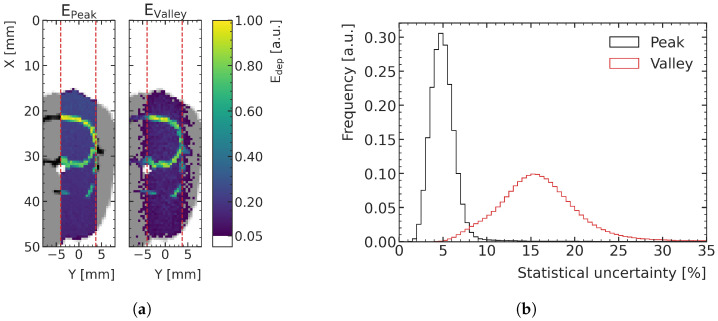
(**a**) Two-dimensional slice of the MC-simulated energy deposition in the peak (**left**) and valley (**right**), respectively, at the center of the prediction volume for an exemplary high-noise training sample (rat number 1), normalized to their respective maximum. The prediction volume is indicated with red dashed lines. Air is shown white, tissue (water) in gray and bone in black. (**b**) Histograms of the voxelwise statistical uncertainties (quantified with the standard error) of the peak and valley energy deposition MC simulations in the high-noise datasets.

**Figure 4 cancers-15-02137-f004:**
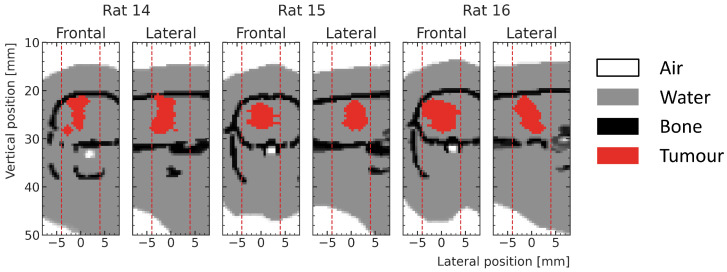
Frontal and lateral slices at the center of the ML prediction regions (red dotted lines) showing exemplary tumors (red) in the respective phantoms (white—air, gray—water and black—bone) of the three test rats used in the testing.

**Figure 5 cancers-15-02137-f005:**
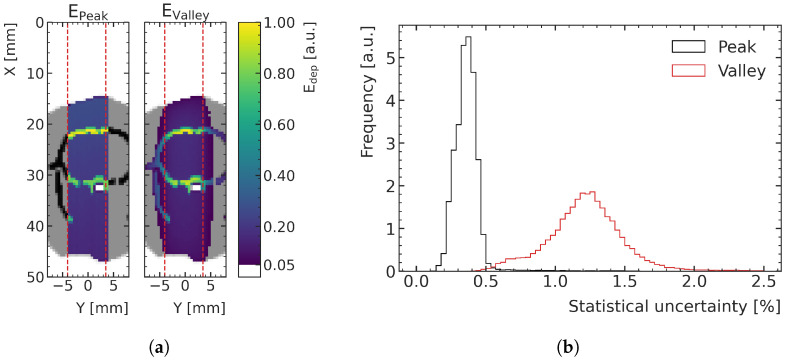
(**a**) Two-dimensional slice of the MC-simulated energy deposition in the peak (**left**) and valley (**right**), respectively, at the center of the prediction volume for an exemplary low-noise training sample (rat number 15), normalized to their respective maximums. Air is shown in white, tissue (water) in gray and bone in black. (**b**) Histograms of the voxelwise statistical uncertainties (quantified with the standard error) of the peak and valley energy deposition MC simulations of the three low-noise treatment test data samples.

**Figure 6 cancers-15-02137-f006:**
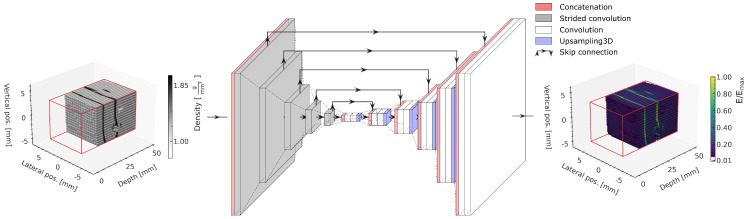
Schematic of the implemented deep learning model predicting energy deposition based on a material matrix input. Adapted from [22].

**Figure 7 cancers-15-02137-f007:**
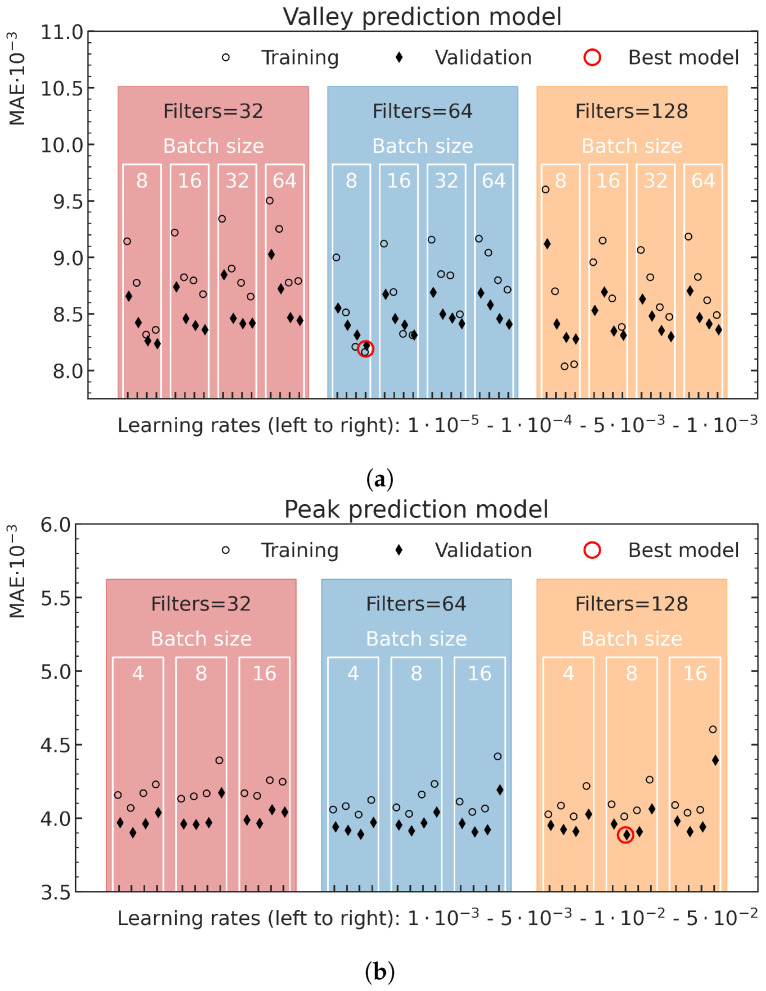
Overview of the validation loss values (diamonds) and the corresponding training data loss values (open circles) for different valley (**a**) and peak (**b**) energy deposition prediction model configurations. The x-ticks locate the different investigated learning rates, while the batch sizes and number of filters are highlighted for each model by their positioning in the respective white (batch size) and colored (number of filters) boxes. The best respective model is marked with a red circle.

**Figure 8 cancers-15-02137-f008:**
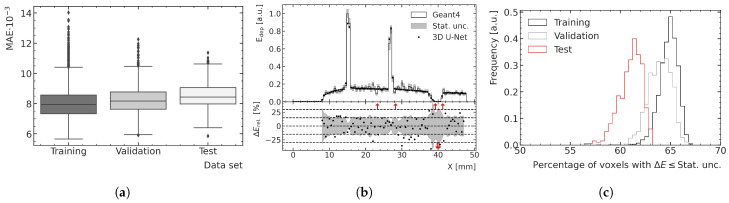
(**a**) Boxplots showing the MAE in the valley region for the training, validation and test datasets. The central line of the each boxplot shows the median of the distribution. The surrounding box is limited by the 25% percentile. The whiskers are shown at the 2.5 × 25% percentile. Data samples further away from the median are represented as outliers. (**b**) Exemplary ML-predicted and MC-simulated energy deposition $E_{\mathrm{dep}}$ of the validation data in the valley region. The bottom plot shows the percent relative difference $\Delta E_{\mathrm{rel}}$ between ML prediction and MC simulation in terms of energy deposition. Red arrows in the relative energy deviation subplot indicate deviations larger than the shown ranges. (**c**) Fraction of voxels of the ML-predicted energy deposition maps exhibiting a deviation of one standard deviation or less with respect to the mean energy deposition ΔE calculated with the MC simulation.

**Figure 9 cancers-15-02137-f009:**
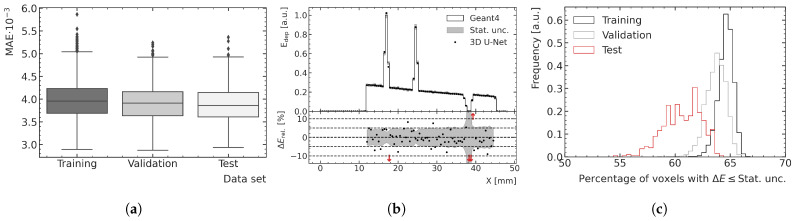
(**a**) Boxplots showing the MAE in the peak region for the training, validation and test datasets. The central line of the each boxplot shows the median of the distribution. The surrounding box is limited by the 25% percentile. The whiskers are shown at the 2.5 × 25% percentile. Data samples further away from the median are represented as outliers. (**b**) Exemplary ML-predicted and MC-simulated energy deposition $E_{\mathrm{dep}}$ of the validation data in the peak region. The bottom plot shows the percent relative difference $\Delta E_{\mathrm{rel}}$ between ML prediction and MC simulation in terms of energy deposition. Red arrows in the relative energy deviation subplot indicate deviations larger than the shown ranges. (**c**) Fraction of voxels of the ML-predicted energy deposition maps exhibiting a deviation of one standard deviation or less with respect to the mean energy deposition ΔE calculated with the MC simulation.

**Figure 10 cancers-15-02137-f010:**
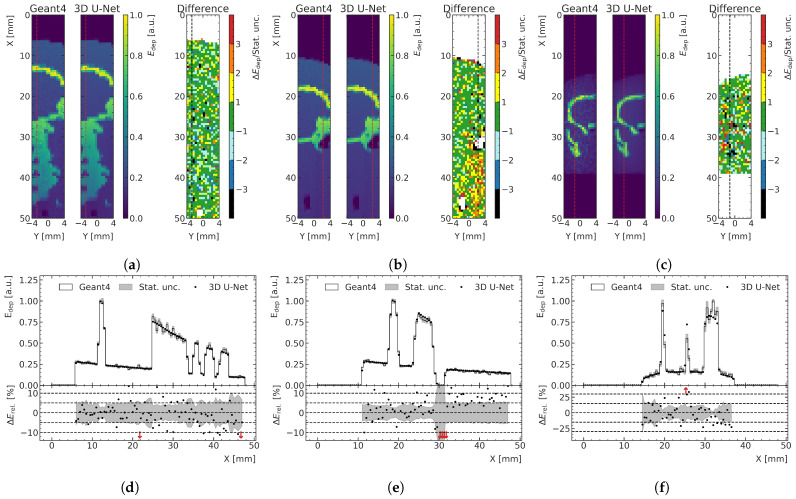
(**a**) Exemplary peak prediction of a training data sample including a larger proportion of spine, showing a 2D slice of MC simulation and ML prediction with the difference in units of statistical standard deviations. (**b**,**c**) show two worst-case prediction cases following different criteria. (**b**) Test data sample with the largest average deviation between ML and MC in units of standard deviations in the peak region. (**c**) Test data sample with the lowest fraction of voxels in which ML prediction with MC simulation agree within one standard deviation in the valley. (**d**–**f**) The depth–energy deposition curve at the position indicated with red (black) dashed line for each 2D representation shown in (**a**–**c**). Red arrows in the relative energy deviation subplot below indicate deviations larger than the shown ranges.

**Figure 11 cancers-15-02137-f011:**
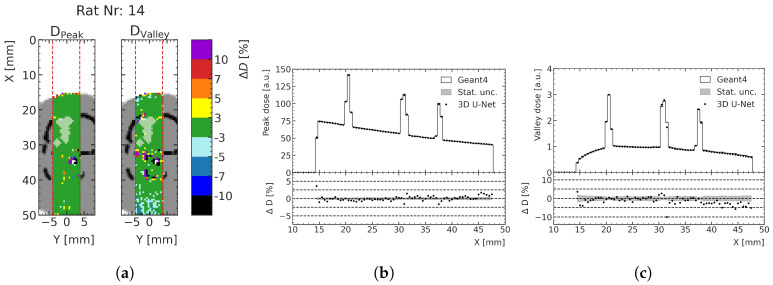
(**a**) Relative dose difference ($\Delta D_{rel}$) between ML and MC models for test rat number 14 in the peak and valley regions. The tumor volume in the shown slice is indicated with a white overlay: (**b**) depth–peak dose curves; (**c**) depth–valley dose curve at the center of the prediction volume. Doses are normalized to the valley doses at the center of the brain.

**Figure 12 cancers-15-02137-f012:**
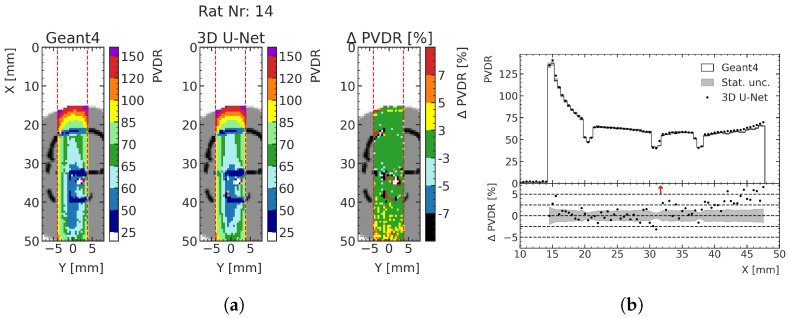
Exemplary comparison of PVDR computed from MC simulation and ML prediction. (**a**) Δ *PVDR* is calculated as $\left(PVDR_{ML}-PVDR_{MC}\right)/PVDR_{ML}$ for each macrovoxel (see Section 2.2). (**b**) Red arrows in the relative PVDR deviation subplot indicate deviations larger than the shown ranges.

**Table 1 cancers-15-02137-t001:** Average MAE and fraction of voxels with an absolute dose difference (Δ*D*) between MC and ML calculation of less than 1σ, computed for the peak and valley predictions. The datasets derive from the MC training, validation and testing. The mean MAEs and associated standard errors are calculated from the MAEs obtained for the augmented data of the 16 rats (10 for training, 3 for validation and 3 for testing). For the voxel fractions (second and forth columns), the reported mean values and standard deviations are calculated considering all the datasets used in the various cases under study, determining the mean value and standard deviation from the individual distributions of each dataset represented in Figure 8c and Figure 9c for an exemplary case.

	Valley		Peak	
Dataset	MAE [$1\times 10^{-3}$]	Δ*D* < Stat. unc. [%]	MAE [$1\times 10^{-3}$]	Δ*D* < Stat. unc. [%]
Training	8.2±0.3	64.8±0.9	4.0±0.2	64.6±0.7
Validation	8.2±0.2	63.9±1.2	3.9±0.1	63.7±0.9
Test	8.4±0.1	61.0±1.1	4.1±0.1	60.7±1.7

**Table 2 cancers-15-02137-t002:** Fraction of voxels with a relative deviation of dose $\Delta D_{rel}$ between ML prediction and low-noise MC simulation of less than 3% in the peak and valley regions, shown for the full phantom, only tissue parts of the phantom (the bone voxels were considered) and the treated tumor volumes.

Rat ID	Peak/Valley	Voxel Ratio with $\Delta D_{\mathrm{rel}}$ < 3% [%]		
		**Full Phantom**	**Tissue Only**	**Tumor Volume**
14	Peak	93.9	95.0	100.0
	Valley	77.6	81.0	95.9
15	Peak	93.9	95.7	100.0
	Valley	81.1	85.0	97.7
16	Peak	94.6	96.1	100.0
	Valley	80.1	83.8	97.9

**Table 3 cancers-15-02137-t003:** Deviation in delivered dose (ΔD) due to a difference in predicted minimum valley dose between ML prediction and MC simulation in the entire prediction region (8×8 cm${}^{2}$ field size).

	Rat 14	Rat 15	Rat 16
Δ *D* [%]	1.17	0.95	−0.37

## Data Availability

Agreements for making the data can be negotiated upon request.

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
