# Peer review of "Accurate and Fast Deep Learning Dose Prediction for a Preclinical Microbeam Radiation Therapy Study Using Low-Statistics Monte Carlo Simulations"

_cancers, 2023, doi:10.3390/cancers15072137_

Round 1
Reviewer 1 Report
Mentzel F et al., presented the work entitled as "Accurate and fast deep learning dose prediction for a preclinical microbeam radiation therapy study using low-statistics Monte Carlo simulations". The work seems to be interesting but I feel there is some issue in method section which needs better clarity.
I will suggest the authors to either minimize the figures in method section or shift the figures and the related text (explanation) need to be shifted in the relevant result section and place.
I must request the authors to write such "1 · 10−3 " in better form in standard way.
Almost all figure labelings are blurred, please think about them write properly. Currently it looks like the labels generated by the codes/software was used as it is. The figures quality needs improvement everywhere.
why in figure legends ":" was used after every subfigure numbers such as (a):???
Figure 8b, 9b, 10d, 10e, and 10f needs detailed legend. Current legend is too short.
In conclusion, I feel the authors need to write more professional english language
Author Response
Dear reviewer,
Thank you very much for taking your time to review our manuscript. We highly value your comments on how to improve the quality of our manuscript. In the following, we discuss each of your remarks and explain how we have modified our manuscript accordingly. Our replies are marked in red colour.
- I will suggest the authors to either minimize the figures in method section or shift the figures and the related text (explanation) need to be shifted in the relevant result section and place.
- We moved several figures to match the text more conveniently. Given the comments provided by other reviewers, the number of figures has been reduced. Resulting from those actions, we think that the issue reported by the reviewer has been solved with the new version of the manuscript.
- I must request the authors to write such "1 · 10−3 " in better form in standard way.
- We have updated the formatting of numbers expressed with scientific notation to match recently published articles in MDPI Cancers. All occurrences changed to “1x10-3” in Table 1.
- Almost all figure labelings are blurred, please think about them write properly. Currently it looks like the labels generated by the codes/software was used as it is. The figures quality needs improvement everywhere.
- The figures provided during manuscript submission were designed to meet the journal requirements (300 dpi for all diagrams). In this revised version of the manuscript, we double-checked that the images are not blurred and we enlarged diagrams where appropriate.
- why in figure legends ":" was used after every subfigure numbers such as (a):???
- We have updated the formatting of subfigure numbers to match recently published articles in MDPI Cancers. All occurrences changed to “(a)” as requested.
- Figure 8b, 9b, 10d, 10e, and 10f needs detailed legend. Current legend is too short.
- We have added extra text in the figures’ captions to provide a clearer interpretation of the plot contents.The captions have been revised for Figures 1, 3, 8, 9, 10, 11 and 12.
- In conclusion, I feel the authors need to write more professional english language
- We have reviewed the entire manuscript, paying special attention to the description of the core concepts, data analysis and conclusions presented in the paper.
Thank you again for your review and insights for the improvement of our manuscript. We hope you find our amendments and explanations satisfactory.
Best regards.
The authors
Reviewer 2 Report
The authors predict and compare dose distributions for non-clinical models of MRT using Monte Carlo simulation and machine learning methods.
In this paper, only the cited references are shown regarding the data calculation method, and it is not possible to understand this paper alone. The authors should add explanations about the following items that can be understood by simply reading this paper.
・Monte Carlo simulation using Geant4
・Explanation of PSF
・Calculation method of MAE
・Stat. Unit calculation method
・ PVDR calculation method and clinically desirable conditions
・Table 2 _ ΔD calculation method
In addition, only the number of rats used is shown, and the extent of sample variability is not known at all. Data such as age in weeks, body weight, distribution of body length and head size, type of tumor, number of days after transplantation and volume should be described.
Furthermore, using 16 rats, 10 rats were used for training, 3 rats for verification, and 3 rats for test data. It is desirable to show justification for the expanded number of data.
Author Response
Dear reviewer,
Thank you very much for taking your time to review our manuscript. We highly value your comments on how to improve the quality of our manuscript. In the following, we discuss each of your remarks and explain how we have modified our manuscript accordingly. Our replies to each comment are marked in red for improved clarity.
- In this paper, only the cited references are shown regarding the data calculation method, and it is not possible to understand this paper alone. The authors should add explanations about the following items that can be understood by simply reading this paper.
- Monte Carlo simulation using Geant4
- Explanation of PSF
- Calculation method of MAE
- Stat. Unit calculation method
- PVDR calculation method and clinically desirable conditions
- Table 2 _ ΔD calculation method
We thank the reviewer for suggestions to strengthen the background knowledge required to read our manuscript.
Regarding:
(i) and (ii)): We added some clarifications in Section 2.1 (lines:88-102)(iii): The formula has been added at line 199, Section 2.6
(iv): The method is now explained at lines 121-124
(v): We have added the following text to the Introduction in lines 60-61. “It also is understood that maximising the peak-to-valley dose ratio (PVDR) results in better biological outcomes [17], thus, accurate estimation of both peak and valley doses is necessary. How the PVDR is calculated is now explained at lines 116-117.
(vi): This is now explained at lines 235-236.
- In addition, only the number of rats used is shown, and the extent of sample variability is not known at all. Data such as age in weeks, body weight, distribution of body length and head size, type of tumor, number of days after transplantation and volume should be described.
- Details have been added at lines 126-131.
- Details have been added at lines 126-131.
- Furthermore, using 16 rats, 10 rats were used for training 3 rats for verification, and 3 rats for test data. It is desirable to show justification for the expanded number of data.
- The ML dose engine training would benefit thousands of individual CT scans, and unfortunately we do not have this number of data available. Therefore, our solution consisted in using the largest number of rats we had (10 rats) for training and to augment the CT scans to be able to train the ML-dose engine with approximately 4,500 “synthetic” CT scans. The CT scans of 3 rats for verification were augmented to ~1,400 and the ones for validation to ~500.
The first paragraph of Section 2.4 has been revised to explain how the available CT data were used (lines 142-148)
- The ML dose engine training would benefit thousands of individual CT scans, and unfortunately we do not have this number of data available. Therefore, our solution consisted in using the largest number of rats we had (10 rats) for training and to augment the CT scans to be able to train the ML-dose engine with approximately 4,500 “synthetic” CT scans. The CT scans of 3 rats for verification were augmented to ~1,400 and the ones for validation to ~500.
Thank you again for your review and insights for the improvement of our manuscript. We hope you find our amendments and explanations satisfactory.
Best regards.
The authors
Reviewer 3 Report
In this study, the authors present and evaluate a Machine Learning (ML) model for predicting dose distributions in preclinical Microbeam Radiotherapy (MRT) applications. Ten rats were used for training the ML model, 3 more for validation/optimization and another 3 for testing. The main characteristic of this model is that training relies on high-noise Monte Carlo (MC) datasets, which are faster to produce. My comments are given below.
1. From a methodological point of view, the study is well-designed and executed. However, I still cannot see the advantage of employing high-noise MC data for training, as compared to using low-noise (high-fidelity) MC data, apart from the (obvious) computational time saved to produce the data. What is the dosimetric benefit in accuracy if low-noise data are used for training? Is it significant? Is it clinically significant for such applications? This should at least be discussed and an example may be presented. If low-noise data are used for training, how much the accuracy of the ML model prediction may increase?
2. The manuscript contains 13 figures, each one with several sub-figures. Presenting such a large number of Figures is unnecessary and confusing to the reader. I highly recommend to remove several (redundant) figures in order to more clearly illustrate the key findings of this study. Figures with redundant information or repeating points already made (in the text or in other figures) should be removed. In my opinion, Figures 1, 4, 6 and 13 can be safely removed without compromising text clarity. However, it is up to the authors to decide which figures to remove and adapt the text accordingly.
3. Furthermore, each figure contains too many sub-figures. As a result, the sub-figures are too small to notice the details and draw conclusions, if the manuscript is printed on a standard paper size. Please reduce the number of sub-figures and increase the size of the remaining sub-figures accordingly. As an example, figures 11 and 13 present results for all three rats used for ML model testing. The authors may consider presenting the results only for one (the most indicative one) rat. Figure 8a and Figure 9a repeat the same results already presented in Table 1.
4. The manuscript should contain a “Simple summary” as described in the guidelines to the authors and also included in the Word template.
5. Lines are not numbered, as in the Word template, and therefore citing specific lines in the text is not practical.
6. Last paragraph of Section 1: “Section 2.1 describes the rat head CT data.” However, Section 2.1 is about the MC simulation. Please correct.
7. The purpose of the red arrows seen in Figures 8, 9, 11 and 12 is not explained in the corresponding captions.
8. Section 3.3, Section 4 (2nd paragraph) and Section 5: the authors use the phrase “nearly all voxels” which is too vague and too general. 77% of the voxels in the valley region is certainly not “nearly all”. Please rephrase and refrain from using such vague statements. Give more quantitative results in the Abstract and Conclusion sections.
9. The Discussion is incomplete, in my opinion. The authors should focus more on the methodological differences between this study and previously published ones. Why results of this study are more accurate, as the authors claim? Please compare the methodologies and highlight any advantages and limitations. What can be improved and how?
10. The authors acknowledge “several limitations” for the presented approach (Discussion, 2nd paragraph). However, the authors mention just two limitations in the text. Please expand this paragraph to include all limitations of this study. E.g., are the three test subjects (numbers 14 to 16) sufficient to evaluate the accuracy of the model? Please justify selection of just three test subjects and discuss.
Author Response
Dear reviewer,
Thank you very much for taking your time to review our manuscript. We highly value your comments on how to improve the quality of our manuscript. In the following, we discuss each of your remarks and explain how we have modified our manuscript accordingly. Our replies to your comments are highlighted red for improved clarity.
- From a methodological point of view, the study is well-designed and executed. However, I still cannot see the advantage of employing high-noise MC data for training, as compared to using low-noise (high-fidelity) MC data, apart from the (obvious) computational time saved to produce the data. What is the dosimetric benefit in accuracy if low-noise data are used for training? Is it significant? Is it clinically significant for such applications? This should at least be discussed and an example may be presented. If low-noise data are used for training, how much the accuracy of the ML model prediction may increase?
- As clearly pointed out by the reviewer, the use of high-noise MC simulations to train the ML model is computationally more effective (the computational time is ~ 2.5% when compared to low noise simulations) . This solution is particularly suitable for MRT because the dose in the valleys is extremely low when compared to the peaks and affected by higher statistical uncertainties when compared to the peaks. A comparison in performance between training with high and low noise MC simulations will be very difficult as we simply do not have a comparable low-noise data set and it is not possible to create one of this type in any realistic time frame. Nevertheless, we believe that this could be a valid study to perform in the future. A statement has been added in the discussion, at lines 409-412.
- As clearly pointed out by the reviewer, the use of high-noise MC simulations to train the ML model is computationally more effective (the computational time is ~ 2.5% when compared to low noise simulations) . This solution is particularly suitable for MRT because the dose in the valleys is extremely low when compared to the peaks and affected by higher statistical uncertainties when compared to the peaks. A comparison in performance between training with high and low noise MC simulations will be very difficult as we simply do not have a comparable low-noise data set and it is not possible to create one of this type in any realistic time frame. Nevertheless, we believe that this could be a valid study to perform in the future. A statement has been added in the discussion, at lines 409-412.
- The manuscript contains 13 figures, each one with several sub-figures. Presenting such a large number of Figures is unnecessary and confusing to the reader. I highly recommend to remove several (redundant) figures in order to more clearly illustrate the key findings of this study. Figures with redundant information or repeating points already made (in the text or in other figures) should be removed. In my opinion, Figures 1, 4, 6 and 13 can be safely removed without compromising text clarity. However, it is up to the authors to decide which figures to remove and adapt the text accordingly.
- Based on this recommendation, we looked again at all figures to retain only those necessary to clearly show our findings. We removed figures: 2a, 11b,c,e,f,h,i, and 13 (completely removed)
- Based on this recommendation, we looked again at all figures to retain only those necessary to clearly show our findings. We removed figures: 2a, 11b,c,e,f,h,i, and 13 (completely removed)
- Furthermore, each figure contains too many sub-figures. As a result, the sub-figures are too small to notice the details and draw conclusions, if the manuscript is printed on a standard paper size. Please reduce the number of sub-figures and increase the size of the remaining sub-figures accordingly. As an example, figures 11 and 13 present results for all three rats used for ML model testing. The authors may consider presenting the results only for one (the most indicative one) rat. Figure 8a and Figure 9a repeat the same results already presented in Table 1.
- The figures have been reduced and some have been enlarged
- The figures have been reduced and some have been enlarged
- The manuscript should contain a “Simple summary” as described in the guidelines to the authors and also included in the Word template.
- Thank you. We added it to the text.
- Thank you. We added it to the text.
- Lines are not numbered, as in the Word template, and therefore citing specific lines in the text is not practical.
- We thank the reviewer for picking up this mistake. We have toggled the line numbers back on for reviewing purposes.
- We thank the reviewer for picking up this mistake. We have toggled the line numbers back on for reviewing purposes.
- Last paragraph of Section 1: “Section 2.1 describes the rat head CT data.” However, Section 2.1 is about the MC simulation. Please correct.
- Thank you for spotting this mistake. We corrected the overview over the contents at the end of the introduction (lines 80-85).
- Thank you for spotting this mistake. We corrected the overview over the contents at the end of the introduction (lines 80-85).
- The purpose of the red arrows seen in Figures 8, 9, 11 and 12 is not explained in the corresponding captions.
- All captions now have the additional following text: “Red arrows in the relative energy deviation subplot below indicate deviations larger than the shown ranges.” This has been clarified in the figures where applicable. The explanation was added to the captions in Figures 8,9,10 (formerly 11) and 12.
- All captions now have the additional following text: “Red arrows in the relative energy deviation subplot below indicate deviations larger than the shown ranges.” This has been clarified in the figures where applicable. The explanation was added to the captions in Figures 8,9,10 (formerly 11) and 12.
- Section 3.3, Section 4 (2nd paragraph) and Section 5: the authors use the phrase “nearly all voxels” which is too vague and too general. 77% of the voxels in the valley region is certainly not “nearly all”. Please rephrase and refrain from using such vague statements. Give more quantitative results in the Abstract and Conclusion sections.
- We have revised all statements that use non quantitative statements and replaced them with the associated numerical values (e.g. lines 318-326)
- We have revised all statements that use non quantitative statements and replaced them with the associated numerical values (e.g. lines 318-326)
- The Discussion is incomplete, in my opinion. The authors should focus more on the methodological differences between this study and previously published ones. Why results of this study are more accurate, as the authors claim? Please compare the methodologies and highlight any advantages and limitations. What can be improved and how?
- In this work we compared the dosimetric accuracy of the ML dose engine against MC simulations, adopted as “gold standard” . We did not compare our engine to other calculations. The advantage of the ML-solution with respect to the other methods published in the literature (refs [25][26]) in this context relies on the quick execution time. We now report a statement on the quick response of the ML dose engine when compared to Monte Carlo simulations in lines 360-366. Another advantage of our method is the adoption of “high-noise” MC simulations to train the ML model, this allows for a significantly faster generation of training datasets. Despite the use of relatively higher noise data, we obtain a satisfactory agreement against MC data in the testing phase. The limitations of the developed ML dose engine are now reported at lines 380-395.
- In this work we compared the dosimetric accuracy of the ML dose engine against MC simulations, adopted as “gold standard” . We did not compare our engine to other calculations. The advantage of the ML-solution with respect to the other methods published in the literature (refs [25][26]) in this context relies on the quick execution time. We now report a statement on the quick response of the ML dose engine when compared to Monte Carlo simulations in lines 360-366. Another advantage of our method is the adoption of “high-noise” MC simulations to train the ML model, this allows for a significantly faster generation of training datasets. Despite the use of relatively higher noise data, we obtain a satisfactory agreement against MC data in the testing phase. The limitations of the developed ML dose engine are now reported at lines 380-395.
- The authors acknowledge “several limitations” for the presented approach (Discussion, 2nd paragraph). However, the authors mention just two limitations in the text. Please expand this paragraph to include all limitations of this study. E.g., are the three test subjects (numbers 14 to 16) sufficient to evaluate the accuracy of the model? Please justify selection of just three test subjects and discuss.
- We have improved the discussion of the limitations of the developed ML dose engine. We believe they are now reported in greater detail, especially regarding the selection of the three test subjects, including a comment on this in the methods section already.
As we have only a very limited number of animal CT scans from this cohort available for our study, we selected the largest group of them to be used as training subjects. The three test rats were chosen arbitrarily as the rats were numbered randomly and not according to any scheme as well. Therefore, we believe that the choice of the three rats is valid although we acknowledge that in future studies our findings should be validated and repeated using a larger number of data points.
- We have improved the discussion of the limitations of the developed ML dose engine. We believe they are now reported in greater detail, especially regarding the selection of the three test subjects, including a comment on this in the methods section already.
Thank you again for your review and insights for the improvement of our manuscript. We hope you find our amendments and explanations satisfactory.
Best regards.
The authors
Round 2
Reviewer 2 Report
Necessary revision have been made.